# AdvQDet: Detecting Query-Based Adversarial Attacks with Adversarial Contrastive Prompt Tuning

## ABSTRACT

Deep neural networks (DNNs) are known to be vulnerable to adversarial attacks even under a black-box setting where the adversary can only query the model. Particularly, query-based black-box adversarial attacks estimate adversarial gradients based on the returned probability vectors of the target model for a sequence of queries. During this process, the queries made to the target model are intermediate adversarial examples crafted at the previous attack step, which share high similarities in the pixel space. Motivated by this observation, stateful detection methods have been proposed to detect and reject query-based attacks. While demonstrating promising results, these methods either have been evaded by more advanced attacks or suffer from low efficiency in terms of the number of shots (queries) required to detect different attacks. Arguably, the key challenge here is to assign high similarity scores for any two intermediate adversarial examples perturbed from the same image. To address this challenge, we propose a novel *Adversarial Contrastive Prompt Tuning* (ACPT) method to robustly fine-tune the CLIP image encoder to extract similar embeddings for any two intermediate adversarial queries. With ACPT, we further introduce a detection framework AdvQDet that can detect 7 state-of-the-art query-based attacks with > 99% detection rate within 5 shots. We also show that ACPT is robust to 3 types of adaptive attacks.

## CCS CONCEPTS

• **Computing methodologies → Machine learning**.

## KEYWORDS

Adversarial example detection, query-based adversarial attacks, adversarial contrastive prompt tuning

## 1 INTRODUCTION

In the past decade, deep neural networks (DNNs) have made remarkable achievements across a wide range of fields, such as computer vision [16, 22], natural language processing [14, 50], and multimodal learning [43, 44]. Despite these advancements, studies have shown that DNNs are extremely vulnerable to small adversarial perturbations at the inference stage [48], which are input perturbations generated to maximize the prediction error of the model. The adversarially perturbed inputs are known as adversarial examples (attacks) and the weakness of DNNs to adversarial attacks is known

*MM '24, 28 October - 1 November 2024, Melbourne, Australia*
© 2024 Copyright held by the owner/author(s). Publication rights licensed to ACM.
ACM ISBN 978-1-4503-XXXX-X/18/06
https://doi.org/XXXXXXX.XXXXXXX

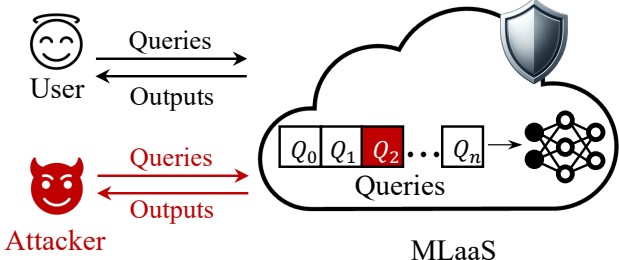

**Figure 1: Query-based attack and stateful detection.**

as the *adversarial vulnerability*. This has raised serious security concerns on the development of DNNs in safety-critical scenarios, such as autonomous driving [4, 17] and medial diagnosis [20, 35].

An adversary could generate adversarial attacks in either a white-box or a black-box setting according to the threat model. In the white-box setting, the adversary has full access to the model's parameters and thus can directly compute the adversarial gradients to generate adversarial examples [36, 48]. In the black-box setting, however, the adversary can only query the target model to estimate the adversarial gradients based on the model returns (probability vectors or hard labels) [3, 8, 57]. Black-box attacks can also be achieved by transfer-based attacks, i.e., generating the attacks based on a surrogate model that is similar to the target model and then applying the generated adversarial examples to attack the target model [7, 15, 51]. Compared to white-box attacks, black-box attacks pose a more practical threat as most commercial models are kept secret from the users except their APIs. In this work, we focus on query-based black-box adversarial attacks and study the detectability of the malicious queries made by these attacks to the target model.

Existing defense approaches against adversarial attacks can be categorized into adversarial training methods [36, 55] and adversarial example detection methods [34, 52]. Although adversarial training has been demonstrated to be one of the most effective defense methods against white-box attacks, it relies on an expensive min-max training of the model. This reduces its utility on large models as even standard training could cost millions of dollars [46]. Adversarial example detection methods, on the other hand, were mostly developed for white-box attacks and thus cannot be applied to detect the intermediate queries made by a black-box adversary to the target model.

One inherent weakness of query-based attacks is that they have to query the target model many times with similar (and partially adversarial) examples generated during the attack process. And those similar queries may be easily detected and rejected by the defender, ideally making the attack fail at the first few attempts. This is known as the *stateful detection* against black-box attacks

[9, 13, 31]. As depicted in Fig.1, by maintaining a list of historical queries, stateful detection works to find the most similar historical query to the current query to determine whether the current query is an adversarial example. If the similarity exceeds a certain threshold, then the current query is detected as an adversarial example. Here, the length of the list introduces a tradeoff between defense effectiveness and storage cost, i.e., a longer list will make the defense more reliable and the attack more expensive but incurs more storage (for each user).

As pixel space detection is sensitive to non-adversarial transformations (e.g., rotation and translation), Chen et al. [9] proposed to leverage a pre-trained CNN to extract features and compare the mean feature similarity between the current query and the last 50 queries from the same user to identify potential attacks. This method can be easily bypassed by Sybil attacks in which the adversary creates multiple fake accounts to evade detection. The Blacklight detection method [31] computes the feature similarity (according to the hamming distance) between the current query and each of the historical queries from all users, and detects if any similarity score is above a certain threshold. Blacklight is thus robust to Sybil attacks. However, it has been shown that existing stateful detection methods all suffer from a poor tradeoff between the detection rate and false positive rate [24], i.e., their thresholds set for high detection rate tend to cause a high false positive rate. This will greatly harm the experience of benign users. Furthermore, the above detection methods have been bypassed by an adaptive attack that attempts to generate dissimilar queries using adaptive step sizes [19].

Arguably, the key to reliable detection of query-based attacks is training a robust feature extractor that always produces similar feature vectors for any two adversarial queries crafted from the same image, even for adaptive attacks. In light of this, we propose a simple yet effective framework, *Adversarial Contrastive Prompt Tuning* (ACPT), to train reliable feature extractors for accurate and robust detection of query-based attacks. Specifically, ACPT finetunes the CLIP image encoder on ImageNet via prompt tuning using two types of losses: 1) contrastive losses to pull together the representations of a clean image and all its adversarial counterparts under data augmentations, and 2) adversarial losses to make it robust to adaptive attacks. Although only finetuned on ImageNet, ACPT demonstrates superb zero-shot capability and achieves the best detection performance across a wide range of datasets.

In summary, our main contributions are:

- We propose a novel *Adversarial Contrastive Prompt Tuning (ACPT)* framework that can train robust feature extractors for stateful detection of query-based attacks.
- We conduct extensive experiments on 5 benchmark datasets against 7 query-based attacks, and show that ACPT can achieve an average 97% and 99% detection rates under 3-shot and 5-shot detection, surpassing the best baseline by > 48% and > 49%, respectively.
- We also show that ACPT is robust to adaptive attacks created by either plugging in an adaptive strategy to existing attacks or a new adaptive strategy that exploits the CLIP image encoder backbone to evade the detection.

## 2 RELATED WORK

Here, we briefly review related works on query-based attacks and stateful detection. We also review existing adversarial contractive learning techniques which are closely related to our adversarial contraction prompt tuning approach.

**Query-based Attacks.** These attacks query the target model repetitively with adversarial examples generated at intermediate steps to obtain more information to enhance the attack. Based on the return type of the target model, query-based attacks can be categorized into score-based attacks (the target model returns confidence scores) and decision-based attacks (the target model returns category labels). The zeroth order optimization (ZOO) [8] attack is one classic score-based attack that exploits finite difference to estimate the adversarial gradients. Compared to ZOO, the autoencoder-based ZOOM (AutoZOOM) [49] attack effectively lowers the average query count required to find successful adversarial examples. Ilyas et al. [25] explored a variant of Natural Evolutionary Strategies (NES) to estimate the adversarial gradient under more restrictive threat models. Andriushchenko et al. [1] further introduced a set of query-efficient score-based black-box attack methods, Square attack, using a randomized search scheme.

For decision-based attacks, the confidence scores are no longer accessible to the adversary, which can only use the label information as a substitute. The Boundary attack [3] and the label-only version of the NES attack [25] are pioneering works in this field. Cheng et al. [12] proposed a novel OPT approach to formulate decision-based attacks as real-valued optimization problems. By using the gradients sign rather than the raw gradients, Cheng et al. [11] further introduced a query-efficient Sign-OPT method to overcome the query limitations faced by all query-based attacks. Another notable method HopSkipJumpAttack (HSJA) [6] employs unbiased gradient estimation at the decision boundary to make the attack more efficient. Following this, an array of decision-based attacks, such as QEBA [32] and SurFree [37], have been developed to reduce the number of queries required to attack unseen DNNs, or decrease the maximum allowed perturbation strength [5].

Table 1: A summary of different stateful detection methods.

| Method | Encoder | Metric | Action |
|--------|---------|--------|--------|
| SD | CNN Encoder | $L_2$ Norm | Ban Account |
| Blacklight | Pixel-SHA | Hamming | Reject Query |
| PIHA | Percept. Hash | Hamming | Reject Query |
| **Ours** | **ACPT** | **Cosine** | **Return Cache** |

**Stateful Detection.** The intuition behind stateful detection is the fact that query-based attacks need to query the target model many times with highly similar queries, as part of the exploration process to find successful adversarial examples. It is thus expected that malicious queries with high similarities can be easily detected in either the pixel or representation space. The stateful detection (SD) method introduced in [9] was the first to examine the users' historical queries to detect query-based attacks. Specifically, SD first extracts the feature of the current query (e.g., an image) using an image encoder and then computes the $L_2$ distance between the query

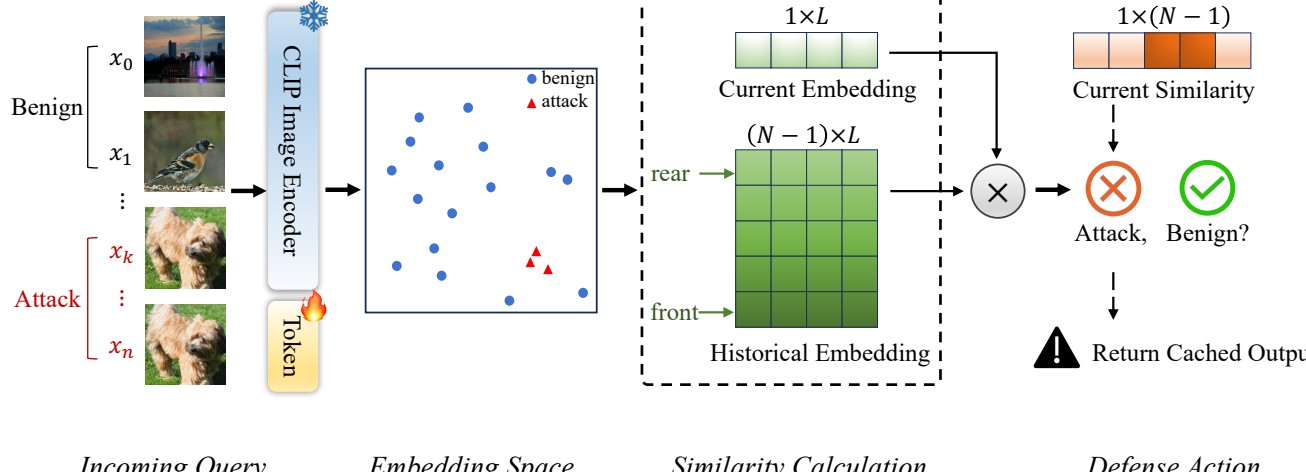

*Incoming Query*          *Embedding Space*          *Similarity Calculation*          *Defense Action*

**Figure 2: An overview of our proposed AdvQDet framework. The current query (e.g., an image) is compared in the embedding space of the CLIP image encoder (finetuned by our ACPT method) with all past queries to detect whether there exists a similar historical embedding. Once the query is detected as an attack (i.e., a similar historical embedding is found), a cashed output from its last queries can be directly returned to avoid returning new information to the adversary.**

feature and its $k$-nearest neighbors found in historical queries of the same user. SD is not robust to Sybil attacks where the adversary creates many fake accounts to distribute the queries and evade user-wise detection. Unlike SD, the Blacklight [31] detection method replaces the feature extractor with the Pixel-SHA probabilistic hash function, which calculates the hash representation for the input image. By further creating a global hash-table to store the historical queries of all users, it establishes a lightweight detection module that can efficiently address the problem of Sybil attacks. Based on Blacklight, PIHA [13] adopts the perceptual image hashing scheme as its feature extractor. It has been shown that stateful detection is also effective against model extraction attacks, which also require a large number of queries to the target model. For example, the PRADA [28] method detects model extraction attacks by analyzing the distribution of consecutive API queries from a user and its deviation from a Gaussian distribution. The SEAT [56] method acquires a similarity encoder via adversarial training, which enables the identification of accounts conducting model extraction attacks. A summary of these methods can be found in Table 1.

**Adversarial Contrastive Learning.** Contrastive learning (CL) [10, 21, 40] is a self-supervised representation learning technique that leverages large-scale unlabeled datasets to train powerful feature extractors. Recently, the concept of adversarial contrastive learning (ACL) [18, 23, 26, 29, 33, 53, 54], has been explored as a robust representation learning technique to combine adversarial training with contrastive learning. Inspired by SimCLR [10], Jiang et al. [26] introduced an unsupervised robust pre-training framework that effectively combines adversarial learning with contrastive pre-training. To avoid implicit knowledge of invariance caused by static augmentation, Dynamic Adversarial Contrastive Learning (DYNACL) [33] employs a dynamic augmentation schedule to bridge the gap between training and test data distributions.

Xu et al. [53] further incorporated causal reasoning and robustness-aware coreset selection (RCS) to help interpret ACL and improve its performance.

## 3 PROPOSED DETECTION FRAMEWORK

We first describe our threat model, formulate the detection problem, and then introduce the proposed ACPT framework that finetunes a robust feature extractor for embedding extraction and similarity calculation. Finally, we devise detection strategies based on the similarity scores and discuss possible defense actions against the detected attack queries.

### 3.1 Threat Model

In this work, we assume a query-based black-box threat model where the adversary generates adversarial examples to attack a target model by making multiple queries to the model and using the model returns to optimize the adversarial examples in an iterative manner. Here, the defender is the owner of the target model who can deploy any defense strategies to defend against potential attacks. In this work, we focus on detection-based defense, which can be deployed in parallel with other defense strategies. However, the defender does not know which user is the attacker nor when the malicious query will arrive. Therefore, the defender may have to store a large number of historical queries of all users to allow a long-range detection of malicious queries. As such, there exists a tradeoff between query storage and detection range. The goal of the defender is to detect any query-based attacks within a minimum number of attempts by the attacker, which forms a few-shot detection setting. There may also exist adaptive attacks that exploit adaptive strategies to evade the detection.

**Figure 3: Our proposed ACPT method. It finetunes the CLIP image encoder using two contrastive losses defined on cleanly and adversarially paired images obtained from the same clean image via data augmentation followed by the PGD attack.**

## 3.2 Problem Formulation

We denote $f_\theta(\mathbf{x}) \rightarrow y$ as a DNN parameterized by $\theta$, where $\mathbf{x} \in \mathcal{X}$ is a clean sample and $y \in \mathcal{Y}$ is its ground-truth label. In image classification tasks, $\mathbf{x}$ represents a clean image and $y \in \{y_1, y_2, \ldots, y_k\}$ is its categorization label; whereas in image captioning tasks, $\mathbf{x}$ is a clean image and $y$ is its associated caption. Given a clean sample $\mathbf{x} \in [0,1]^d$ and a target model $f_\theta(\cdot)$, a query-based adversarial attack aims to generate an adversarial example $\mathbf{x}'$ that maximizes the loss of the model as follows:

$$\mathbf{x}' = \underset{\|\mathbf{x}'-\mathbf{x}\|_\infty \leq \epsilon}{\arg\max} \; \ell(f(\mathbf{x}'), y), \tag{1}$$

where $\ell(\cdot)$ is the loss function, $\mathbf{x}'$ is an intermediate-step adversarial example, and $\epsilon$ is the perturbation budget. An adversarial attack can either be untargeted as formulated above or targeted toward a target label $y'$. Please note that our work does not differentiate between targeted and untargeted attacks.

A query-based black-box attack solves the above adversarial optimization problem by estimating the adversarial gradients iteratively as follows:

$$\mathbf{x}'_{t+1} = \mathbf{x}'_t + \eta \, \text{sign}(\hat{g}), \tag{2}$$

where $\mathbf{x}'_t$ is the intermediate adversarial example obtained at the $t$-th iteration, $\eta$ is the perturbation step size, $\text{sign}(\cdot)$ is the sign function, and $\hat{g}$ is the estimated gradient based on target model output $f(\mathbf{x}'_t)$ using a black-box optimization method such as finite difference [8] or NES [25].

For a current query $\mathbf{x}_t$, the task of stateful detection is to determine whether there exists a historical query $\mathbf{x}_k$ such that their similarity exceeds a certain threshold $\mu$. Formally, it is:

$$det(\mathbf{x}_t) = \begin{cases} 1, & \text{if } sim(E(\mathbf{x}_t), E(\mathbf{x}_k)) > \mu, \exists \mathbf{x}_k \in Q \\ 0, & \text{otherwise,} \end{cases} \tag{3}$$

where $det(\cdot)$ is the detection function, $sim(\cdot, \cdot)$ is the similarity function, $E(\cdot)$ is an encoder (feature extractor) that extracts the embedding of $\mathbf{x}_t/\mathbf{x}_k$, $\mu$ is a threshold hyper-parameter, and $Q$ is an embedding bank that stores the embeddings of historical queries from all users. Here, a $det(\cdot)$ value of 1 indicates an attack. Note

that $E(\cdot)$ is a different model from the target model $f(\cdot)$ and is an adversarially finetuned CLIP [43] image encoder by our ACPT method.

## 3.3 AdvQDet Framework

*3.3.1 Overview.* As illustrated in Figure 2, AdvQDet consists of 2 main components: 1) the ACPT finetuned image encoder and 2) a similarity calculation module. The detection procedure of AdvQDet is as follows. For a current query $\mathbf{x}_t$, it first feeds the image into the ACPT finetuned image encoder to extract its embedding. The similarity calculation module then compares the embedding with $N-1$ historical embeddings of the past queries (from all users) to compute the similarity scores. If any of the $N-1$ similarity scores say $\mathbf{x}_k$ is above a pre-defined threshold $\mu$, query $\mathbf{x}_t$ will be determined as a potential attack. Instead of rejecting $\mathbf{x}_t$, one plausible defense action is to just return the cached output for $\mathbf{x}_k$. Note that existing detection methods employ two types of strategies for embedding bank $Q$. The SD method [9] creates a local bank for each user, while later methods Blacklight [31] and PIHA [13] maintain a global bank for all users. Our AdvQDet also adopts the global bank strategy as it is robust to Sybil attacks. Next, we will introduce the two components in detail.

*3.3.2 Adversarial Contrastive Prompt Tuning.* As depicted in Figure 3, ACPT adopts a two-stream contrastive prompt tuning paradigm [26]: a *clean stream* and a *adversarial stream*. In the clean stream, the two augmented views (e.g., $\tilde{\mathbf{x}}_i$ and $\tilde{\mathbf{x}}_j$) of a clean image $\mathbf{x}$ form a Clean-to-Clean (C2C) pair. The purpose of the clean stream is to pull together the augmented versions of the same image, making it robust to different types of image transformations. In the adversarial stream, the adversarial examples of the two augmented images are generated using PGD [36] to form an Adversarial-to-Adversarial (A2A) pair. The purpose of the adversarial stream is to make it robust to adaptive attacks that exploit adversarial perturbation to bypass the detection. Together, the two streams robustify the image encoder against both regular transformations and adversarial

perturbations. Note that the clean stream itself is the standard SimCLR [10].

To exploit the superb feature extraction capability of large-scale pre-trained models, we adopt the image encoder of CLIP [43] and apply ACPT to finetune the encoder on ImageNet. ACPT adopts visual prompt tuning with learnable prompt tokens concatenated to the original input tokens. The tuning loss of ACPT is defined as follows:

$$\ell_{NT}(\tilde{\mathbf{x}}_i, \tilde{\mathbf{x}}_j; p) = -\log \frac{\exp(sim(E(\tilde{\mathbf{x}}_i, p), E(\tilde{\mathbf{x}}_j, p))/\tau)}{\sum_{k=1}^{2N} \exp(sim(E(\tilde{\mathbf{x}}_i, p), E(\tilde{\mathbf{x}}_k, p))/\tau)}, \quad (4)$$

$$\ell_{ANT}(\tilde{\mathbf{x}}_i', \tilde{\mathbf{x}}_j'; p) = -\log \frac{\exp(sim(E(\tilde{\mathbf{x}}_i', p), E(\tilde{\mathbf{x}}_j', p))/\tau)}{\sum_{k=1}^{2N} \exp(sim(E(\tilde{\mathbf{x}}_i', p), E(\tilde{\mathbf{x}}_k', p))/\tau)}, \quad (5)$$

$$\ell_{\text{ACPT}} = \alpha \ell_{NT}(\tilde{\mathbf{x}}_i, \tilde{\mathbf{x}}_j; p) + (1 - \alpha)\ell_{ANT}(\tilde{\mathbf{x}}_i', \tilde{\mathbf{x}}_j'; p), \quad (6)$$

where $p$ is the prompt token, $E(\cdot)$ is the CLIP image encoder, $sim(\cdot, \cdot)$ is the cosine similarity function, $\tau$ is the temperature, and $\alpha = 0.5$ is a hyperparameter balancing the two loss terms.

Comparing the definition of $\ell_{\text{ACPT}}$ and Eq. (3), one might find that $\ell_{\text{ACPT}}$ directly optimizes the feature similarity between the clean and adversarial image pairs. This effectively reduces the difference between variants of the same image in the latent space, making the detection of query attacks much easier.

*3.3.3 Similarity Calculation.* Following prior works [13, 31], we extract and save the embedding of each query image into an embedding bank $Q$. The embedding bank is maintained globally for all users so as to be robust to Sybil attacks. Two problems arise with the embedding bank: 1) the storage cost and 2) the computational cost. The two costs can be reduced by using the techniques introduced in [9]. Next, we will provide an analysis of the two costs and show that it is practically feasible to store a global embedding bank and perform similarity search efficiently.

In terms of the storage cost, each query results in a vector embedding with dimension $d = 512$, which takes 2048 bytes for float32 precision. Suppose there are 1 million users with each user querying 100 times, the storage it takes to store all these query embeddings is 190.73 GB. By switching to float16 precision, the storage can be reduced to 95.37 GB.

In terms of computational cost, one can use the Automatic Mixed Precision (AMP) technique to reduce the memory cost and accelerate computations without sacrificing the detection performance. AMP automatically determines the appropriate precision—single or half—for each operation. When calculating the cosine similarity between an individual embedding vector and each embedding in the embedding bank, the computational complexity is $O(n \times d)$, where $n$ is the number of embeddings in the bank and $d$ is the dimension of the embedding vector. There are established techniques we can use to speed up high-dimensional similarity search, such as product quantization (PQ), hierarchical navigable small worlds (HNSW), and locality sensitive hashing (LSH). Popular similarity search tools like clip retrieval [2], Faiss [27], and AutoFaiss all provide efficient solutions for searching over a large-scale vector database. Here, we conduct an efficiency test to compute the cosine similarity between two vectors of dimensions $(1, 512)$ and $(1m, 512)$ using an NVIDIA RTX 3090 GPU, CUDA 11.3, and Pytorch v1.12.0.

It takes 8.29 and 2.63 milliseconds for float32 and float16, respectively. These costs are manageable for an AI company to run a commercial product/service that supports up to 1 million users.

*3.3.4 Defense Action.* Once a query is detected to be an attack, there are a few possible defense actions that can be taken by the defender: 1) rejecting the query, which is applicable when the false positive rate is low as otherwise may harm user experience; 2) limiting the query number and frequency of the user which will cause the attacker's attention; 3) returned intentionally perturbed outputs to the user which still has the risk to leak gradient (or other) information; 4) banning accounts or blocking IP addresses which is an aggressive action that should be taken only in extreme cases; and 5) simply returning the cashed output for the previous similar query which is a plausible action that does not expose new information to the user nor harm the user experience.

## 4 EXPERIMENTS

We evaluated our detection method against 7 state-of-the-art query-based attacks and 3 types of adaptive attacks. We first describe our experimental setting and then present the results of 1) defense effectiveness across different datasets, 2) robustness to adaptive attacks, and 3) ablation study.

### 4.1 Experimental Setup

**Datasets and Models.** We experiment on 5 benchmark datasets: CIFAR-10 [30], GTSRB [47], ImageNet [45], Flowers [39], Pets [42]. We utilize ImageNet pre-trained models (such as ResNet20, ResNet101, and ViT-B/16) and then fine-tune them on the other four datasets. A summary of these datasets and the corresponding models can be found in the Appendix.

**Attack Configuration.** We evaluate against 7 query-based attacks, including Boundary [3], HSJA [6], NESS [25], QEBA [32], Square [1], SurFree [37], and ZOO [8], as described in Section §2. We also apply an adaptive strategy called Oracle-guided Adaptive Rejection Sampling (OARS) [19] to enhance the above query-based attacks and evaluate against these enhanced attacks. OARS utilizes an adapting distribution and resampling technique for gradient estimation, aiming to evade stateful defenses during the generation of adversarial examples. Throughout the experiment, we execute each attack until an adversarial example is successfully crafted or the maximum query limit is reached, whichever occurs first. The hyperparameters for these attacks are set following the Adversarial-Robustness-Toolbox(ART) library [38]. For the attacks, we set the perturbation budget to $\epsilon = 0.05$ and limit the query budget to 100, 000. For CIFAR-10 and GTSRB datasets, we randomly choose 1,000 images from their respective test sets, uniformly across all categories. For ImageNet, Flowers, and Pets datasets, due to the high computational costs of query-based attacks, we select 100 images randomly from the validation/test sets.

**Defense Configuration.** For existing stateful detection methods, we use their originally proposed configurations, as detailed in Table 1. Specifically, for SD [9] defense, we set the number of neighbors to $k = 50$ and the detection threshold to $\mu = 10$. For Blacklight [31], the quantization step is set to 50, with window sizes of 20 for CIFAR-10 and 50 for ImageNet. PIHA [13] adopts a block size of 7x7 and a detection threshold of $\mu = 0.05$.

Table 2: The ASR (↓), 3/5-shot detection rate (↑) and mean detection counts (↓) of different detection methods against 7 query-based attacks across 5 datasets. The best and second best results are boldfaced and underscored, respectively.

| Dataset | Attack Method | Stateful Detection Method | | | | | | | | | | |
| | | w/o Defense | | Blacklight | | | PIHA | | | AdvQDet (Ours) | | |
| | | ASR | Query | ASR | 3/5-shot DR | mDC | ASR | 3/5-shot DR | mDC | ASR | 3/5-shot DR | mDC |
|---|---|---|---|---|---|---|---|---|---|---|---|---|
| CIFAR-10 | Boundary | 100% | 591.97 | 0% | 94%/97% | 3.23 | 0% | 75%/93% | 3.87 | **0%** | **100%/100%** | **3.00** |
| | HSJA | 100% | 265.11 | 0% | 0%/0% | 7.28 | 0% | 1%/14% | 7.77 | **0%** | **76%/100%** | **2.90** |
| | NESS | 100% | 15144.82 | 0% | **100%/100%** | 3.00 | 0% | 89%/97% | 3.64 | **0%** | 98%/98% | **2.81** |
| | QEBA | 100% | 316.41 | 0% | 0%/0% | 7.28 | 0% | 1%/14% | 7.77 | **0%** | **76%/100%** | **2.90** |
| | Square | 100% | 17.37 | 0% | 100%/100% | 2.00 | 28% | 61%/64% | 2.96 | **0%** | **100%/100%** | **2.00** |
| | SurFree | 100% | 77.13 | 0% | 0%/0% | 8.66 | 0% | 3%/10% | 8.85 | **0%** | **100%/100%** | **2.00** |
| | ZOO | 71% | 16649.93 | 0% | 100%/100% | 2.00 | 0% | 100%/100% | 2.00 | **0%** | **100%/100%** | **2.00** |
| ImageNet | Boundary | 100% | 5776.94 | 4% | 16%/19% | 238.21 | 8% | 0%/0% | 228.88 | **0%** | **100%/100%** | **3.00** |
| | HSJA | 74% | 79621.63 | 0% | 0%/0% | 8.51 | 0% | 0%/1% | 9.56 | **0%** | **83%/100%** | **3.86** |
| | NESS | 99% | 13276.7 | 0% | **100%/100%** | 3.07 | 10% | 19%/21% | 266.88 | **0%** | 99%/100% | **2.51** |
| | QEBA | 59% | 55173.28 | 0% | 0%/0% | 8.51 | 0% | 0%/1% | 9.56 | **0%** | **83%/100%** | **3.86** |
| | Square | 100% | 108.2 | 0% | 100%/100% | 2.00 | 30% | 22%/24% | 9.1 | **0%** | **100%/100%** | **2.00** |
| | SurFree | 100% | 534.95 | 0% | 0%/0% | 9.02 | 0% | 0%/1% | 9.68 | **0%** | **100%/100%** | **2.04** |
| | ZOO | 75% | 9986.08 | 0% | 100%/100% | 2.00 | 0% | 99%/99% | 4.26 | **0%** | **100%/100%** | **2.00** |
| GTSRB | Boundary | 100% | 1908.37 | 0% | 100%/100% | 3.03 | 0% | 81%/93% | 3.97 | **0%** | **100%/100%** | **3.00** |
| | HSJA | 100% | 1808.87 | 0% | 0%/0% | 7.29 | 0% | 11%/56% | 6.47 | **0%** | **100%/100%** | **2.56** |
| | NESS | 49% | 51501.31 | 0% | **100%/100%** | **3.00** | 0% | 50%/77% | 5.16 | **0%** | 95%/96% | 4.84 |
| | QEBA | 100% | 780.26 | 0% | 0%/0% | 7.29 | 0% | 11%/56% | 6.47 | **0%** | **100%/100%** | **2.58** |
| | Square | 100% | 2577.15 | 0% | 100%/100% | 2.00 | 7% | 71%/71% | 3.68 | **0%** | **100%/100%** | **2.00** |
| | SurFree | 75% | 225.77 | 0% | 0%/5% | 7.84 | 0% | 16%/51% | 6.56 | **0%** | **100%/100%** | **2.00** |
| | ZOO | 42% | 18708.50 | 0% | 100%/100% | 2.00 | 0% | 100%/100% | 2.00 | **0%** | **100%/100%** | **2.00** |
| Flowers | Boundary | 96% | 5118.87 | 15% | 6%/9% | 297.24 | 25% | 0%/0% | 375.63 | **0%** | **100%/100%** | **3.00** |
| | HSJA | 56% | 59574.49 | 0% | 0%/0% | 8.67 | 0% | 0%/0% | 9.26 | **0%** | **99%/100%** | **3.77** |
| | NESS | 95% | 17092.08 | 0% | **100%/100%** | 3.01 | 6% | 53%/64% | 101.58 | **0%** | 99%/99% | **2.56** |
| | QEBA | 100% | 54968.15 | 0% | 0%/0% | 8.67 | 0% | 0%/0% | 9.26 | **0%** | **99%/100%** | **3.77** |
| | Square | 99% | 324.59 | 0% | 100%/100% | 2.00 | 29% | 48%/50% | 5.49 | **0%** | **100%/100%** | **2.00** |
| | SurFree | 99% | 1704.45 | 0% | 0%/0% | 9.98 | 0% | 0%/0% | 10.71 | **0%** | **100%/100%** | **2.00** |
| | ZOO | 87% | 9197.09 | 0% | 100%/100% | 2.00 | 0% | 98%/99% | 2.07 | **0%** | **100%/100%** | **2.00** |
| Pets | Boundary | 95% | 7958.55 | 3% | 16%/18% | 245.19 | 2% | 0%/0% | 197.13 | **0%** | **100%/100%** | **3.00** |
| | HSJA | 97% | 2277.19 | 0% | 0%/0% | 8.61 | 0% | 0%/1% | 9.45 | **0%** | **100%/100%** | **3.61** |
| | NESS | 94% | 23424.64 | 0% | 100%/100% | 3.07 | 12% | 6%/10% | 425.60 | **0%** | **100%/100%** | **2.00** |
| | QEBA | 97% | 1061.13 | 0% | 0%/0% | 8.61 | 0% | 0%/1% | 9.45 | **0%** | **100%/100%** | **3.61** |
| | Square | 100% | 148.85 | 0% | 100%/100% | 2.00 | 8% | 14%/14% | 10.45 | **0%** | **100%/100%** | **2.00** |
| | SurFree | 100% | 754.22 | 0% | 0%/0% | 10.88 | 0% | 0%/2% | 11.08 | **0%** | **100%/100%** | **2.03** |
| | ZOO | 86% | 7919.80 | 0% | 100%/100% | 2.00 | 0% | 100%/100% | 2.00 | **0%** | **100%/100%** | **2.00** |
| **Average** | | 90% | 17671.10 | 1% | 49%/50% | 27.12 | 5% | 32%/39% | 51.03 | **0%** | **97%/99%** | **2.66** |

**Implementation Details.** For our AdvQDet, we finetune the CLIP image encoder using ACPT for 20 epochs with a batch size of $bs = 1024$ and a learning rate of 0.04 on ImageNet. To generate a batch of positive pairs for finetuning, we sample $bs$ images from the training set and then follow SimCLR to obtain two augmented views $(\tilde{x}_i, \tilde{x}_j)$. We apply PGD attack to craft the adversarial views $(\tilde{x}'_i, \tilde{x}'_j)$ with a perturbation budget of 8/255 for 5 steps. After obtaining the four views $(\tilde{x}'_i, \tilde{x}'_j, \tilde{x}_i, \tilde{x}_j)$, we fine-tune the prompt token by minimizing the adversarial contrastive loss described in Section §3.3.2. There are $K = 20$ learnable prompt tokens, optimized by SGD and adjusted by cosine annealing. For detection, we set a similarity threshold of $\mu = 0.95$ for low-resolution datasets CIFAR-10 and GTSRB, and $\mu = 0.9$ for high-resolution datasets ImageNet, FLowers, and Pets.

**Performance Metrics.** We consider three performance metrics: 1) attack success rate (ASR), which is the percentage of successful adversarial examples under the attack budget; 2) 3/5-shots (queries) detection rate (DR) which is the successful detection rate when the defender sees 3/5 of the queries (i.e., a clean query followed by a sequence of adversarial queries), and 3) mean detection counts (mDC) which calculates the average number of queries required for the defender to detect each attack.

**Table 3: The ASR($\downarrow$) and mean detection counts ($\uparrow$) of different detection methods against 6 enhanced query-based attacks by the OARS adaptive strategy. The results are shown for CIFAR-10 and ImageNet datasets with the best results are boldfaced.**

| Dataset | Attack Method | Stateful Detection Method | | | | | | | | | |
|---|---|---|---|---|---|---|---|---|---|---|---|
| | | w/o defense | | SD | | Blacklight | | PIHA | | **AdvQDet** | |
| | | ASR | Query | ASR | mDC | ASR | mDC | ASR | mDC | ASR | mDC |
| CIFAR-10 | Boundary-OARS | 100% | 610.31 | 100% | 51.00 | 100% | 3.17 | 94% | 4.01 | **0%** | **3.00** |
| | HSJA-OARS | 100% | 439.78 | 100% | 51.00 | 100% | 7.28 | 93% | 7.77 | **0%** | **2.90** |
| | NESS-OARS | 100% | 969.14 | 53% | 51.00 | 97% | 596.50 | 97% | 381.78 | **0%** | **3.00** |
| | QEBA-OARS | 100% | 457.14 | 100% | 51.00 | 98% | 7.28 | 93% | 7.77 | **0%** | **2.90** |
| | Square-OARS | 100% | 183.64 | 100% | 51.00 | 98% | 64.94 | 100% | 83.85 | **0%** | **3.20** |
| | SurFree-OARS | 100% | 170.52 | 65% | 51.41 | 92% | 8.66 | 61% | 8.85 | **0%** | **2.00** |
| ImageNet | Boundary-OARS | 100% | 5743.65 | N/A | N/A | 37% | 194.75 | 39% | 208.10 | **0%** | **3.00** |
| | HSJA-OARS | 100% | 1908.77 | N/A | N/A | 93% | 9.00 | 98% | 9.56 | **0%** | **3.86** |
| | NESS-OARS | 100% | 5207.24 | N/A | N/A | 89% | 282.51 | 55% | 428.31 | **0%** | **3.01** |
| | QEBA-OARS | 100% | 1040.41 | N/A | N/A | 73% | 8.51 | 100% | 11.00 | **0%** | **3.86** |
| | Square-OARS | 99% | 840.77 | N/A | N/A | 83% | 40.88 | 99% | 70.87 | **0%** | **2.53** |
| | SurFree-OARS | 100% | 1519.29 | N/A | N/A | 87% | 9.02 | 100% | 9.68 | **0%** | **2.04** |
| **Average** | | 99% | 1590.89 | 86.33% | 51.07% | 87.25 | 102.71 | 85.75% | 102.63 | **0%** | **2.94** |

## 4.2 Main Results

We compare our AdvQDet method with 7 existing stateful detection methods. For a fair comparison, we adopt the same defense pipline for all methods. I.e., we detect each query based on the historical queries from all users, with only the similarity score is computed by different detection methods. The detection performance results are reported in Table 2, where the 3-4 columns report the results of no defense. It is evident that, although most attacks can achieve a high ASR (nearly 100%) in the absence of detection, they often require a large number of queries to succeed. According to the results, the Square attack is the most efficient and effective attacks as it requires the minimum number of queries and achieves an ASR of 100% across all datasets.

For the detection methods, our AdvQDet achieves the best average performance of 0% ASR, 97%/99% 3/5-shot detection rate, and an average of 2.66 query counts for successful detection, surpassing existing methods Blacklight and PIHA by a huge margin. Moreover, AdvQDet demonstrates the best performance and almost 100% 3/5-shot detection rates in most scenarios. However, it is not always the best, for example, the Blacklight detection method works better against the NESS attack than AdvQDet in terms of 3/5-shot detection rates. This is because NESS attack uses a large Gaussian noise distribution to estimate the adversarial gradients which tend to cause large distortion to the query images and thus the features. However, Blacklight extracts the hashing of the image which is relatively robust to large perturbations. However, Blacklight fails badly against HSJA, QEBA, and SurFree attacks with almost 0% 3/5-shot detection rates. It is worth mentioning that AdvQDet is very close to Blacklight against NESS but can detect the attacks within fewer queries.

Although query-based attacks generally require many queries while detention only needs a few queries, there are still attacks that can bypass existing detection methods Blacklight and PIHA. For example, the Square, NESS, and Boundary attacks on high

resolution datasets ImageNet, Flowers, and Pets. By contrast, not a single existing query-based attack can evade our detection, leaving an ASR of 0% in all scenarios. Efficiency is another advantage of our AdvQDet method, i.e., it only takes 2.66 queries on average to detect all 7 attacks. Note that, the first query made by most attacks is a clean image, the second query is often an initialized image with Gaussian noise, and the third query is an adversarial query. This means that our method can detect most of the attacks based on the first two queries, for example, against HSJA and QEBA attacks.

## 4.3 Robustness Against Adaptive Attacks

Here, we evaluate the robustness of our method to adaptive attacks where the attackers are aware of our detection pipeline. Particularly, we consider three adaptive attacks: 1) using OARS [19] adaptive strategy to boost existing attacks; 2) the attacker knows the backbone (CLIP image encoder) of our AdvQDet; and 3) white-box attacks where the attacker knows every detail of our detector (but the target model is still black-box).

**OARS Adaptive Attack.** OARS employs step size adaptation and resampling mechanisms to evade stateful detection. We boost existing attacks including Boundary, HSJA, NESS, QEBA, Square, and SurFree using the OARS adaptive strategy. We did not consider the ZOO attack as its adaptive strategy is not compatible with OARS and it is also omitted from the OARS paper [19]. The robustness results on CIFAR-10 and ImageNet datasets are shown in Table 3. It is clear that when there is no defense, all adaptive attacks achieve an ASR of $\geq$ 99 with the query number increase significantly on high resolution images (ImageNet).

Our AdvQDet is robust to OARS adaptive attacks and can successfully detect all 6 adaptive attacks within an average of 3 shots while reducing the ASR to 0%. The SD detection method however fails on ImageNet as its feature extractor is dataset-dependent and thus is not applicable to ImageNet images. Since SD requires the last 50 queries to detect the current, the mean detection counts are

all above 50. The Blacklight and PIHA have both been bypassed by all adaptive attacks, where the ASR jumps up to 37% - 100%. Interestingly, Blacklight is more susceptible to adaptive attacks on low-resolution dataset CIFAR-10 while PIHA is more vulnerable on both low and high-resolution datasets CIFAR-10 and ImageNet.

**The Backbone is Compromised.** Here, we test when the attacker knows the CLIP image encoder used in AdvQDet (but not the visual prompt token). In this case, the attacker can white-box attack the CLIP image encoder while query attacking the target model. Specifically, the attacker adopts an alternating optimize strategy to first perform one step (query) black-box attack and then 10 steps of white-box PGD attack. As shown in Figure 4, AdvQDet is also robust to this adaptive attack, maintaining a high similarity score for the first 50 steps of queries. Moreover, AdvQDet becomes more robust when we increase the token length of ACPT.

**White-box Attack.** In this case, we follow a similar adaptive pipeline as in the above backbone adaptive attack setting, but the attacker directly attacks our ACPT-tuned image encoder. The results are also presented in Figure 4. The result indicates that AdvQDet is moderately robust to white-box attacks with a slightly reduced similarity score, and increasing the token length of ACPT can effectively increase the chance of the attack being detected. Note that in both experiments, the detection is deemed to be successful whenever the similarity score is above the threshold which occurs within the first 5 queries. We also observed that white-box attacks against our AdvQDet took roughly 100x more queries to converge. These results suggest that with ACPT, we can have a reliable query attack detector with good effectiveness, efficiency, and robustness.

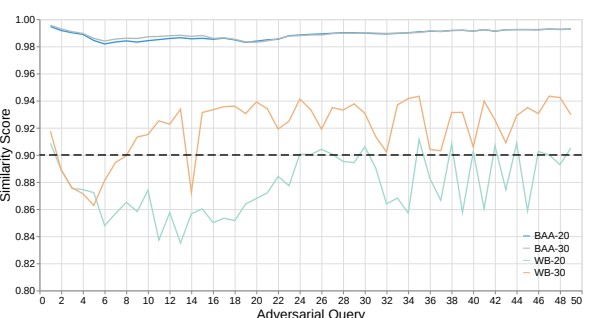

**Figure 4: The similarity score of the first 50 queries for backbone adaptive attacks ("BAA-x") and white-box attacks ("WB-x") on ImageNet, with x denoting the token length. The black dashed line marks the detection threshold.**

### 4.4 Further Analysis

**Effect of Prompt Token Length.** Here, we analyze the impact of prompt token length of ACPT on the detection performance, with varying token lengths $K \in [0, 30]$. Note that when $K = 0$, the ACPT-tuned encoder degenerates to the vanilla CLIP image encoder. As depicted in Figure 5, our AdvQDet can reliably distinguish between benign and adversarial queries, assigning high average similarity scores (close to 1 almost everywhere) to adversarial queries. The difference is more pronounced as the token length of ACPT increases.

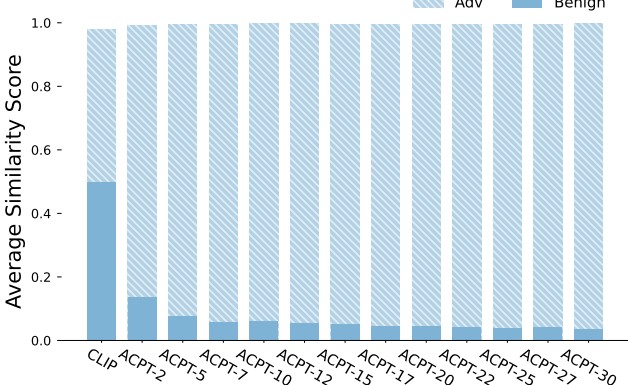

**Figure 5: The average similarity score of the first 50 benign and adversarial queries under varying prompt token length ("ACPT-x" with x denoting the token length) on ImageNet.**

## 5 LIMITATION

As a stateful detection method, our AdvQDet also faces certain limitations that deserve further research. Notably, it cannot defend against transfer-based attacks as they do not need querying the target model. This limitation can potentially be addressed by incorporating white-box adversarial example detection methods into the pipeline of AdvQDet. The storage and computational costs mark another limitation of AdvQDet. More effective partitioning and acceleration techniques can be developed in future work to facilitate the industrial deployment of AdvQDet. On the other hand, besides its effectiveness, efficiency, and robustness, AdvQDet has the potential to be applied to detect multimodal query-based attacks against vision language models (VLMs) like GPT-4V [41]. Although there is still much room for improvement, we believe AdvQDet offers a reliable solution for detecting query-based adversarial attacks.

## 6 CONCLUSION

In this paper, we proposed a novel stateful detection framework to detect query-based black-box adversarial attacks. Our work is motivated by the observation that query-based attacks launch multiple visually similar queries to the target model, which might be easily detected by a robust feature extractor (image encoder). To this end, we propose an efficient tuning-based method called *Adversarial Contrastive Prompt Tuning* (ACPT) to robustify the CLIP image encoder on ImageNet. The ACPT-tuned serves as a general-purpose encoder for the detection of query-based attacks, and demonstrates strong zero-shot generalization capability across different datasets. With ACPT, we introduce the AdvQDet framework that extracts and saves the embeddings of the query images and maintains a global embedding bank for all users. AdvQDet computes the embedding similarity between the current query and all historical queries to identify whether the query is malicious (similar to an existing one). We demonstrated the effectiveness, efficiency, and robustness of AdvQDet against existing query-based attacks, adaptive attacks, and even white-box attacks. Our work showcases the possibility of achieving strong and consistent defense against query-based adversarial attacks.

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
