# OpenReview forum: "AdvQDet: Detecting Query-Based Adversarial Attacks with Adversarial Contrastive Prompt Tuning"
_acmmm.org/ACMMM/2024/Conference — MM2024 Poster_

### Official Review · Reviewer_AUcC · 2024-05-19

**Rating:** 4
**Confidence:** 2

**Summary:**

The paper employs the CLIP model as an image encoder and integrates adversarial contrastive prompt tuning to enhance the discriminative performance of adversary embeddings. The results indicate superior performance compared to other stateful detection methods using simpler encoders.

**Strengths:**

* The paper introduces a novel stateful detection scheme that leverages the CLIP encoder combined with contrast learning, showcasing a robust approach to defense against query-based adversarial attacks.

* The evaluation of existing attacks and defensive baselines demonstrate the effectiveness of the proposed method.

**Limitations:**

* The paper does not explore alternative adversarial sample optimization strategies beyond PGD in the context of Adversarial Contrastive Prompt Tuning, raising concerns about the generality of the proposed method.

* It is not clear whether the CLIP model requires ACPT for each dataset as well as whether the benefits observed on high-resolution datasets (e.g., ImageNet) could effectively transform to lower-resolution datasets.

* The explanation of "white-box attacks" in Section 4.3 is vague, particularly in how an attacker uses knowledge of the CLIP image encoder.

* The benefits of contrast training on adversarial samples appear trivial as depicted in Figure 5. Given the high threshold (e.g., 0.9 on ImageNet), the original CLIP encoder might already sufficiently classify adversarial samples without additional training.

**Suitability:**

2

---

### Official Review · Reviewer_5ZUp · 2024-05-21

**Rating:** 3
**Confidence:** 2

**Summary:**

This paper proposes a defense against query-based adversarial attacks that utilize similar queries to estimate local gradients. The method involves using a CLIP image encoder to embed each incoming query. By comparing these embeddings to historical ones, the system returns the cached output of the query whose embedding closely matches the incoming query's embedding, thereby preventing new information leakage to the attacker.

**Strengths:**

1. This paper is well-written, and each concept is clearly explained.
2. The proposed ACPT method effectively detects various query-based attacks across different benchmark datasets, surpassing the state-of-the-art (SOTA) baseline.

**Limitations:**

1. **Limited Novelty:** The proposed Adversarial Contrastive Prompt Tuning (ACPT), which aims to train robust feature extractors, is highly similar to existing adversarial contrastive learning methods[18, 23, 26, 29, 33, 53, 54]. Both approaches share the core idea of combining adversarial training with contrastive learning.

2. **Underexplained Key Insights:** The role of contrastive vision prompt tuning and its advantages and disadvantages compared to other contrastive learning methods are not sufficiently explained. For instance, the integration of the CLIP encoder may limit the embedding power of the image encoder due to the necessity to align with the text encoder. Additionally, if the adversarially trained image encoder is already robust to adversarial inputs, why not use it directly for classification? Maintaining an embedding encoder and historical embeddings incurs additional costs, but it is unclear how much this enhances robustness.

3. **Computational Cost:** The training cost of adversarial contrastive learning is not provided. Does this method share the same computational drawbacks as adversarial training, as mentioned in lines 101-105?

4. **Missing Details of the Adaptive Attack:** There is a lack of information about the adaptive step size and the reasons for the adaptive attack's failure.

**Suitability:**

2

---

### Official Review · Reviewer_nCJ1 · 2024-05-25

**Rating:** 4
**Confidence:** 3

**Summary:**

The manuscript introduces the Adversarial Contrastive Prompt Tuning (ACPT), a method designed to enhance the robustness of feature extraction against query-based black-box adversarial attacks. By fine-tuning the CLIP image encoder with contrastive and adversarial losses, ACPT generates similar embeddings for adversarial queries derived from the same image, demonstrating improved detection capabilities in stateful detection systems.

**Strengths:**

1. The paper tackles a significant and timely challenge in adversarial attack detection by developing a feature extractor capable of reliably identifying similar adversarial queries.
2. The ACPT framework shows exceptional detection rates, significantly outperforming existing methods on multiple benchmarks, as evidenced by extensive experiments demonstrating up to 99% detection accuracy within 5 shots across diverse datasets.
3. The framework is robust against adaptive attacks, designed specifically to evade detection methods, highlighting its potential for practical security applications.

**Limitations:**

1. The paper primarily focuses on query-based attacks, yet the current trend in the field is leaning towards both query- and transfer-based methods, which are not addressed in the experiments. This oversight limits the generalizability and applicability of the proposed method. Relevant studies in this area include:

   [1] Y. Feng, et al., "Boosting black-box attack with partially transferred conditional adversarial distribution," CVPR, 2022.

   [2] J. Liu, et al., "Difattack: Query-efficient blackbox adversarial attack via disentangled feature space," AAAI, 2024.

   [3] F. Yin, et al., "Generalizable black-box adversarial attack with meta learning," IEEE Trans. Pattern Anal. Mach. Intell., 2024.

2. There is a noticeable absence of an ablation study on the hyperparameter $\alpha$, which could provide deeper insights into the sensitivity and tuning of the model.
3. The manuscript lacks experimental results on the storage and computational costs associated with the AdvQDet, which are crucial for evaluating the practical deployment of the proposed framework.

Suggestions for Improvement:
- Expand the scope of the experiments to include both query- and transfer-based adversarial methods to enhance the relevance and applicability of the ACPT framework.
- Include an ablation study focusing on the hyperparameter $\alpha$ to detail its impact on the performance and optimization of the model.
- Provide a detailed analysis of the storage and computational requirements of the AdvQDet system to assess its feasibility for real-world applications.

**Suitability:**

3

---

### Meta-Review · Area_Chair_T7Dq · 2024-07-01

**Recommendation:** Accept (Poster)
**Confidence:** 5

**Metareview:**

The reviewers acknowledged the rebuttal letter, and have merged to similar ratiting for accepting this work. There are still some major concerns about providing more details and explanations about the failure of ACL and the application of the proposed method in zero-shot scenarios, providing more quantitative results to support the conclusion. Thus, I recommend the acceptance as a poster work.